# Growth Differentiation Factor-15 (GDF-15) Is a Biomarker of Muscle Wasting and Renal Dysfunction in Preoperative Cardiovascular Surgery Patients

**DOI:** 10.3390/jcm8101576

**Published:** 2019-10-01

**Authors:** Toshiaki Nakajima, Ikuko Shibasaki, Tatsuya Sawaguchi, Akiko Haruyama, Hiroyuki Kaneda, Takafumi Nakajima, Takaaki Hasegawa, Takuo Arikawa, Syotaro Obi, Masashi Sakuma, Hironaga Ogawa, Shigeru Toyoda, Fumitaka Nakamura, Shichiro Abe, Hirotsugu Fukuda, Teruo Inoue

**Affiliations:** 1Department of Cardiovascular Medicine, School of Medicine, Dokkyo Medical University, Shimotsuga-gun, Tochigi 321-0293, Japan; tswg0814@gmail.com (T.S.); hal@dokkyomed.ac.jp (A.H.); hirokane1010@gmail.com (H.K.); sho-taka.07@softbank.ne.jp (T.N.); thasegawa6134@gmail.com (T.H.); takuoari@dokkyomed.ac.jp (T.A.); syoutarouobi@yahoo.co.jp (S.O.); masakuma@dokkyomed.ac.jp (M.S.); s-toyoda@dokkyomed.ac.jp (S.T.); abenana@dokkyomed.ac.jp (S.A.); inouet@dokkyomed.ac.jp (T.I.); 2Department of Cardiovascular Surgery, School of Medicine, Dokkyo Medical University, Shimotsuga-gun, Tochigi 321-0293, Japan; sibasaki@dokkyomed.ac.jp (I.S.); hironaga_0722@yahoo.co.jp (H.O.); fukuda-h@dokkyomed.ac.jp (H.F.); 3Third Department of Internal Medicine, Teikyo University, Chiba Medical Center, Ichihara, Chiba 299-0111, Japan; fumitaka@med.teikyo-u.ac.jp

**Keywords:** GDF-15, cardiovascular surgery, operative risk, biomarkers, muscle wasting, sarcopenia, renal dysfunction, chronic kidney disease

## Abstract

Frailty and sarcopenia increase the risk of complications and mortality when invasive treatment such as cardiac surgery is performed. Growth differentiation factor-15 (GDF-15) involves various pathophysiological conditions including renal dysfunction, heart failure and cachexia. We investigated the pathophysiological roles of preoperative GDF-15 levels in cardiovascular surgery patients. Preoperative skeletal muscle index (SMI) determined by bioelectrical impedance analysis, hand-grip strength, 4 m gait speed, and anterior thigh muscle thickness (TMth) measured by echocardiography were assessed in 72 patients (average age 69.9 years) who underwent cardiovascular surgery. The preoperative serum GDF-15 concentration was determined by enzyme-linked immunosorbent assay. Circulating GDF-15 level was correlated with age, brain natriuretic peptide, and estimated glomerular filtration rate (eGFR). It was also negatively correlated with SMI, hand-grip strength, and anterior TMth. In multivariate analysis, eGFR and anterior TMth were the independent determinants of GDF-15 concentration even after adjusting for age, sex, and body mass index. Alternatively, the GDF-15 level was an independent determinant of eGFR and anterior TMth. We concluded that preoperative GDF-15 levels reflect muscle wasting as well as renal dysfunction in preoperative cardiovascular surgery patients. GDF-15 may be a novel biomarker for identify high-risk patients with muscle wasting and renal dysfunction before cardiovascular surgery.

## 1. Introduction

As life expectancy has increased, the number of older patients undergoing cardiovascular surgery has also increased, and many such patients have frailty. Frailty is a geriatric syndrome described as decreased reserves when confronted with stressors, which is associated with poor outcomes in both community cohorts and patients [1,2]. In cardiovascular surgery such as transcatheter valve implantation (TAVI), frailty is now identified as an important predictor of adverse outcomes in older patients [1,3,4]. Patients with higher frailty are at increased risk during the postoperative period, with long hospital stays, and postoperative complications such as stroke and death, compared to those with lower frailty [5]. Sarcopenia, the skeletal muscle loss associated with aging, also includes low physical function (hand-grip strength, walking speed) as a component of frailty [6,7]. It is frequently associated with chronic diseases including heart failure, chronic kidney disease (CKD), cancer, wasting, and cachexia [7]. Thus, it is generally accepted that frailty and sarcopenia, as well as physical function are predictors of survival in patients with cardiovascular diseases (CVD), and they increase the risk of complications and mortality when invasive treatments such as cardiovascular operations are performed. Therefore, it is quite important to identify high-risk patients with frailty and sarcopenia before cardiovascular surgery.

Growth differentiation fator-15 (GDF-15) is not highly expressed in most tissues under normal physiological conditions [8,9], but it increases under various pathophysiological conditions such as inflammation [10], oxidant stress [11], and ischemia/reperfusion [12]. Clinical trials reported that GDF-15 is a reliable biomarker of CVD [13,14,15], and heart failure [16], and it has independent prognostic value in patients with coronary artery bypass grafting (CABG) [17], acute coronary syndromes (ACS) [18,19], and heart failure (HF) [20]. In addition, it has been reported that GDF-15 is a novel serum biomarker of mortality in CKD, and can identify patients at high risk of developing CKD [21,22,23]. The preoperative GDF-15 level has been also reported to be a novel risk biomarker in association with the EuroSCORE for risk stratification independently of N-terminal pro-B-type natriuretic peptide (NTproBNP) and high-sensitive troponin T [24], and closely related to post-operative morbidity in cardiovascular surgery patients including those undergoing TAVI [25,26]. Furthermore, it has been reported to reflect post-operative acute kidney injury (AKI) in patients undergoing CABG [27] and myocardial injury in patients undergoing off-pump CABG [28]. Thus, GDF-15 appears to be a novel biomarker to identify surgical risk in patients undergoing cardiovascular surgery.

Circulating GDF-15 levels increase with age [20,22], and may partly reflect mitochondrial dysfunction in aging and age-related diseases [29]. Furthermore, several papers showed that GDF-15 may be involved in muscle wasting in a variety of patients including COPD [30], patients undergoing elective high-risk cardiothoracic surgery [31], intensive care unit (ICU) patients [32], and cancer patients [33]. Bloch et al. [31] showed that an elevation of postoperative GDF-15 levels is a potential factor associated with muscle atrophy in patients undergoing cardiovascular surgery. However, the physiological significance of the preoperative GDF-15 level in patients undergoing cardiovascular surgery still remains to be clarified.

Therefore, we investigated the pathophysiological roles of preoperative GDF-15 levels in cardiovascular surgery patients. We provided the first evidence that GDF-15 may be a novel biomarker for identifying high-risk patients with muscle wasting and renal dysfunction, compared with tumor necrosis factor α (TNFα) or insulin growth factor-1 (IGF-1) in cardiovascular surgery patients.

## 2. Methods

### 2.1. Participants

A total of 72 preoperative patients (42 males, 30 females) undergoing cardiovascular surgery at Dokkyo Medical Hospital were recruited in this study. The patient characteristics are summarized in Table 1. The mean age was 69.9 ± 13.1 years (23–89 years), and body mass index (BMI) was 24.3 ± 3.9 kg/m^2^. Most of the patients had conventional risk factors such as hypertension (HT), diabetes (DM), hyperlipidemia (HL), current smoking, and hemodialysis (HD) as shown in Table 1. The average preoperative New York Heart Association (NYHA) classification was 2.2 ± 1.0. Table 1 also shows the number of patients classified by surgical procedures for cardiovascular disease. The study was approved by the Ethics Committee of the Dokkyo Medical University (No. 27077), and informed consents were obtained from all participants.

The biochemical data were analyzed using routine chemical methods in Dokkyo Medical University Hospital clinical laboratory. Hemoglobin A1c (HbA1c), brain natriuretic peptide (BNP), and estimated glomerular filtration rate (eGFR) were measured before the operation. The eGFR was calculated by the following equations:Males: eGFR (mL/min/1.73m^2^) = 194 (creatinine^−1.094^) (age^−0.287^)
Females: eGFR (mL/min/1.73m^2^) = 0.739 {194(creatinine^−1.094^) (age^−0.287^)}


The high-sensitivity C-reactive protein (hsCRP) was measured by a latex-enhanced nephelometric immunoassay (N Latex CRP II and N Latex SAA, Dade Behring Ltd., Tokyo, Japan). The homeo-static model assessment of insulin resistance (HOMA-IR), which indicates an index of insulin resistance, was calculated from the fasting blood insulin (immunoreactive insulin (IRI)) concentration and the fasting blood glucose (FBS) level early in the morning, based on the following equation.
HOMA-IR = (IRI) (FBS)/405


To measure fasting serum GDF-15, TNFα, and IGF-1 levels, peripheral venous blood was drawn into pyrogen-free tubes with and without EDTA on the morning of cardiovascular surgery. Plasma and serum were stored in aliquots at −80 °C for all enzyme linked immunosorbent assay (ELISA).

### 2.2. Enzyme Linked Immunosorbent Assay (ELISA)

Serum GDF-15 level was measured by a Human Quantikine ELISA Kit (DGD150 for GDF-15, R&D Systems, Minneapolis, MN, USA). Samples, reagents, and buffers were prepared according to the manufacturers’ manuals. The detection threshold of GDF-15 was 2.0 pg/mL. The serum concentrations of TNFα were measured by a Human Quantikine HS ELISA Kit (HSTA00E, R&D Systems, Minneapolis, MN, USA), and the detection threshold was 0.022 pg/mL. The serum IGF-1 concentration was measured by a Human Quantikine ELISA Kit (DG100, R&D Systems), and the detection threshold was 0.026 ng/mL. 

### 2.3. Measurement of Gait Speed, Hand-Grip Strength, and Voluntary Isometric Contraction

Maximum voluntary isometric contraction (MVIC) of the hand-grip was measured by using a factory-calibrated hand dynamometer (TKK 5401, TAKEI Scientific Instruments Co., Ltd., Tokyo, Japan). Each patient performed two trials, and the highest value was adopted for analysis. The gait speed was measured as the time needed to walk 4 m at an ordinary pace. MVIC of the knee extensors was measured by using a digital handheld dynamometer (μTas MT-1, ANIMA Co., Ltd., Tokyo, Japan) as described previously [34]. Each subject performed two trials, and the highest score was adopted for analysis. 

### 2.4. Measurements with the Bioelectrical Impedance Analyzer (BIA)

A multi-frequency bioelectrical impedance analyzer (BIA), InBody S10 Biospace device (Biospacte Co, Ltd., Korea/Model JMW140) was used to measure muscle and fat volume as described in detail previously [34]. Thirty impedance measurements were performed using 6 different frequencies (1, 5, 50, 250, 500, and 1000 kHz) at the five segments of the body (right arm, left arm, trunk, right leg, and left leg). The measurements were performed while the subjects rested in the supine position, with their elbows extended and relaxed along their trunk. Body fat volume, body fat percentage, and skeletal muscle volume were measured. Skeletal muscle mass index (SMI; appendicular skeletal muscle mass/height^2^, kg/m^2^) was also calculated as the sum of lean soft tissue of the two upper limbs and two lower limbs. In this study, sarcopenia was defined according to the Asian Working Group for Sarcopenia (AWGS) [7] criteria (age ≥ 65 years; handgrip < 26 kg or gait speed ≤ 0.8 m/s, and SMI < 7.0 kg/m^2^ for males; handgrip < 18 kg or gait speed ≤ 0.8 m/s, and SMI < 5.7 kg/m^2^ for females. 

### 2.5. Measurement of Muscle Thickness by Ultrasound 

The anterior mid-thigh muscle thickness was measured on the right leg using a real-time linear electronic scanner with a 10.0-MHz scanning head and Ultrasound Probe (L4–12t-RS Probe, GE Healthcare Japan, Tokyo, Japan) and LOGIQ e ultrasound (GE Healthcare Japan), as previously described [34]. From the ultrasonic image, the subcutaneous adipose tissue-muscle interface and the muscle-bone interface were identified. The perpendicular distance from the adipose tissue-muscle interface to the muscle-bone interface was considered to represent the anterior thigh muscle thickness (TMth). The measurement was performed twice in both the supine and standing positions, and the average value was adopted for analysis.

### 2.6. Statistical Analysis

Data are presented as mean value ± SD. After testing for normality (Kolmogorov-Smirnov), the comparison of means between groups was analyzed by a two-sided, unpaired Student’s *t*-test in the case of normally distributed parameters or by the Mann-Whitney-U-Test in the case of non-normally distributed parameters. Associations among parameters were evaluated using Pearson or Spearman correlation coefficients. Multiple linear regression analysis with log (serum GDF-15 concentration), eGFR, or anterior TMth as the dependent variable was performed to identify the independent factors (clinical laboratory data, or physical data) that influenced these dependent variables. Age, sex, and BMI were employed as covariates. When the independent data were not normally distributed, the data were logarithmically transformed to achieve a normal distribution. Receiver operating characteristic (ROC) curves were plotted to identify an optimal cut-off level of the serum concentration of GDF-15 for detecting impaired eGFR. All analyses were performed using SPSS version 24 for Windows (IBM Corp., New York, NY, USA). A *p* value less than 0.05 was regarded as significant.

## 3. Results

### 3.1. Patient Characteristics

All patients had medical treatment including β-blocking agents (49%), calcium-channel blockers (38%), angiotensin receptor blockers (ARB)-/-angiotensin converting enzyme inhibitors (ACEI) (58%), diuretics (49%), statins (53%), and anti-diabetic drugs (31%) (Table 1). 

The sex differences of the study patients are shown in Table 2. The mean age of females was significantly higher than that of males (*p* < 0.05). The BMI value was not different between males and females, but body fat percentage (%) in females was significantly higher, compared with that in men (*p* < 0.05). 

The hand-grip strength, knee extension strength, and SMI in females were significantly lower than those in males. The anterior TMth (standing) in males was significantly higher, compared with that in females (*p* < 0.05). The BNP level was not different between males and females (399 ± 673 pg/mL vs. 268 ± 345 pg/mL). The mean eGFR of all the patients was 58.2 ± 24.0 mL/min/1.73 m^2^. Furthermore, patients were classified into five groups based on the eGFR levels: normal (eGFR ≥ 90 mL/min/1.73 m^2^), low (eGFR 60–89 mL/min/1.73 m^2^), moderate (eGFR 30–59 mL/min/1.73 m^2^), severe (eGFR 15–29 mL/min/1.73 m^2^), and kidney failure (eGFR < 15 mL/min/1.73 m^2^). Among total 72 patients, there were 3 patients (normal), 36 patients (low), 25 patients (moderate), 2 patients (severe), and 6 patients (kidney failure). Therefore, the overall prevalence of CKD (eGFR < 60 mL/min/1.73 m^2^) was 33 patients out of 72 (44%). The serum TNFα level was 3.5 ± 2.8 pg/mL in all of the patients. It was higher in males than in females (males, 4.1 ± 2.9 pg/mL; females, 2.6 ± 1.9 pg/mL, *p* < 0.05). The serum GDF-15 level did not significantly differ between males and females (males, 1928 ± 1655 pg/mL; females, 1325 ± 1078 pg/mL). The serum IGF-1 concentration also did not significantly differ between males and females (males, 77.3 ± 36.5 ng/mL; females, 70.4 ± 28.5 pg/mL).

### 3.2. Correlation between Various Parameters and Serum GDF-15, TNFα, and IGF-1 Concentration

The correlations between serum GDF-15, TNFα, and IGF-1 concentrations and the clinical data are shown in Table 3 and Figure 1. Table 3 shows the data obtained from males and females separately, and Figure 1 shows the correlations in all of the patients. The serum GDF-15 level was positively correlated with age in all of the patients (*r* = 0.438, *p* < 0.001, Figure 1Aa), but not TNFα (*r* = 0.192, *p* = 0.105, Figure 1Ba), and serum IGF-1 concentration declined with age (*r* = −0.346, *p* = 0.003, Figure 1Ca). In addition, the serum GDF-15 level was positively correlated with age in both males and females (males, *r* = 0.436, *p* = 0.004; females, *r* = 0.637, *p* < 0.001). The concentration of GDF-15 was positively correlated with BNP in both sexes (males, *r* = 0.427, *p* = 0.005; females, *r* = 0.480, *p* = 0.007), but the serum TNFα level was positively correlated with BNP only in men (*r* = 0.465, *p* = 0.002). The concentration of GDF-15 and TNFα (Figure 1Ab, Figure 1Bb), but not IGF-1 (Figure 1Cb), was negatively correlated with eGFR (Figure 1Ab, GDF-15, *r* = −0.768, *p* < 0.001; Figure 1Bb, TNFα, *r* = −0.551, *p* < 0.001) in all of the patients and both sexes (Table 2). Both GDF-15 and TNFα were negatively correlated with hemoglobin (Hb) in males (GDF-15, *r* = −0.560, *p* < 0.001; TNFα, *r* = −0.566, *p* < 0.001).

Table 3 also shows the relationships between serum GDF-15, TNFα, IGF-1 concentrations and the physical data. Figure 2 shows the correlations between serum GDF-15, TNFα, IGF-1 concentrations and the physical data in men. The serum GDF-15 level was negatively correlated with hand-grip strength, SMI, anterior TMth (supine, and standing) in both males and females, as shown in Table 3. As shown in Figure 2, there was a negative correlation between the serum GDF-15 level and both anterior TMth (supine) (*r* = −0.636, *p* < 0.001, Figure 2Aa), and hand-grip strength (*r* = −0.456, *p* = 0.002, Figure 2Ab) in men. Similar results were obtained in females as shown in Table 3. On the other hand, a negative correlation was observed between the serum TNFα level and grip strength, knee extension strength, SMI, and anterior TMth (supine, standing) in males, but not in females (Table 3). A negative correlation between the serum TNFα level and both anterior TMth (supine) (*r* = −0.509, *p* = 0.001, Figure 2Ba), and grip strength (*r* = −0.393, *p* = 0.010, Figure 2Bb) was observed in men. On the other hand, a positive correlation was observed between the serum IGF-1 level and anterior TMth (supine) in men (*r* = 0.429, *p* = 0.005, Figure 2Ca, Table 3), but not hand-grip strength (*r* = 0.240, *p* = 0.126, Figure 2Cb, Table 3). Furthermore, there were no correlations between the serum IGF-1 level and anterior TMth (supine, and standing) in females. 

### 3.3. Relationships among Serum GDF-15, TNFα, and IGF-1 Concentration

Table 3 shows the relationships among the serum GDF-15, TNFα, and IGF-1 concentrations. The serum level of GDF-15 was positively correlated with that of TNFα in both sexes (males, r = 0.657, *p* < 0.001; females, *r* = 0.434, *p* = 0.017). A negative correlation was observed between the serum IGF-1 concentration and GDF-15 (*r* = −0.553, *p* = 0.002) and TNFα (*r* = −0.479, *p* = 0.007) in females, but not males.

### 3.4. Multiple Regression Analysis of Serum GDF-15 Levels and the Clinical Parameters

The linear regression analysis with serum GDF-15 levels as the dependent variable and clinical data (eGFR, BNP, Hb, SMI, hand-grip strength and anterior TMth (supine) as independent variable were investigated in all of the patients as shown in Table 4A. Univariate regression analysis (Table 4A) showed that eGFR (β = −0.650, *p* < 0.001), and anterior TMth (supine) (β = −0.358, *p* = 0.001) were independent variable to predict serum GDF-15 levels. Multiple regression analysis also showed that eGFR (β = −0.597, *p <* 0.001) and anterior TMth (supine) (β = −0.272, *p* = 0.019) were the independent determinants of GDF-15 concentration after adjusting for age, sex, and body mass index. 

Alternatively, multiple regression analysis showed that GDF-15 (β = −0.390, *p* = 0.004) was the independent determinant of anterior TMth (supine), even adjusting for age, sex, and BMI (Table 4B).

The regression analysis between eGFR and the clinical data (BNP, CRP, Hb, GDF-15, and TNFα) were performed as shown in Table 4C. Multiple regression analysis showed that GDF-15 (β = −0.565, *p* <0.001) and TNFα (β = −0.197, *p* = 0.050) were the independent variable to predict eGFR after adjusting for age, sex, and BMI. These results suggest that the serum GDF-15 concentration is a predictor for low eGFR. The ROC curves were plotted to identify the optimal cut-off levels of GDF-15, TNFα, and Hb for detecting eGFR lower than 60 mL/min/1.73 m^2^, which was approximately the same value as the mean eGFR (58.2 mL/min/1.73 m^2^). To construct the ROC curves, different cut-off values of GDF-15, TNFα, and Hb were used to predict eGFR lower than 60 mL/min/1.73 m^2^, with true positives plotted on the vertical axis (sensitivity) and false-positives (1-specificity) plotted on the horizontal axis. The area under the curves (AUCs) for GDF-15, TNFα, and Hb were 92.1%, 78.6%, and 67.5%, respectively. Sensitivity and specificity were 88.2% and 82.9% for GDF-15, 81.8% and 61.5% for TNFα, and 45% and 83% for Hb, respectively. The optimal cut-off value of GDF-15 was 1154 pg/mL as shown in Figure 3. 

### 3.5. Relationships between Sarcopenia and Serum Concentration of GDF-15, TNFα and IGF-1

Sarcopenia was identified in 24 (36%) of a total of 66 patients evaluated based on the sarcopenia criteria. Serum GDF-15, TNFα and IGF-1 levels were compared in patients with and without sarcopenia (Table 5). Patients with sarcopenia had significantly higher age and BNP levels in both males and females, compared to those without sarcopenia (Table 5). On the other hand, they had significant lower gait speed, hand-grip strength, extension strength, SMI, and anterior TMth (supine, standing). Furthermore, both males and females with sarcopenia had significantly higher GDF-15 levels, compared to those without sarcopenia (males, 1483 ± 1125 pg/mL vs. 3053 ± 2346 pg/m, *p* < 0.05; females, 891 ± 700 pg/mL vs. 1625 ± 1302 pg/mL, *p* < 0.05). The TNFα concentration was significantly higher in males with than without sarcopenia (3.06 ± 1.95 pg/mL vs. 6.07 ± 3.27 pg/mL, *p* < 0.01). IGF-1 levels did not significantly differ between patients with and without sarcopenia in both males and females.

## 4. Discussion

The major findings of the present study are as follows: (1) In preoperative cardiovascular surgery patients, the circulating GDF-15 level was correlated with age, BNP, and eGFR. It also had a negative correlation with SMI, hand-grip strength, and anterior TMth. The GDF-15 levels were significantly higher in sarcopenia patients. (2) In multivariate analysis, eGFR and anterior TMth were independent determinants of GDF-15 concentration after adjusting for age, sex, and BMI. Furthermore, the GDF-15 level was an independent determinant of eGFR and anterior TMth. These results suggests that an increased GDF-15 level reflects muscle wasting as well as renal dysfunction in preoperative cardiovascular surgery patients. Thus, GDF-15 may be a novel biomarker for identifying high-risk patients with muscle wasting and renal dysfunction in cardiovascular surgery patients. 

### 4.1. Association of Serum GDF-15 Levels with eGFR

The present study showed that the preoperative GDF-15 level in patients undergoing cardiovascular surgery including CABG and AVR was positively associated with age, eGFR, and BNP. This is compatible with the previous paper showing that preoperative GDF-15 levels in patients undergoing CABG were positively associated with age, chronic renal failure, and high NT-proBNP [25]. Clinical studies have also shown that the GDF-15 concentration correlated strongly with age in healthy elderly individuals and patients with coronary artery diseases [20,28]. GDF-15 is expressed in various tissue types such as kidneys, macrophages, cardiomyocytes, and endothelial cells. The expression occurs in response to various stimuli including oxidative and metabolic stress, tissue injury, and inflammation [10,11,12]. It has been reported that an increase in GDF-15 levels in community-based patients is associated with endothelial dysfunction and subclinical cardiovascular disease [35]. 

Moreover, clinical trials have shown that GDF-15 can be regarded as a reliable biomarker of CVD [13,14,15], and chronic heart failure [16]. In addition, GDF-15 is a novel serum biomarker of mortality in CKD and can identify patients at high risk of developing CKD [21,22]. The presence and progression of CVD are often intimately associated with CKD [36,37]. Valvular heart disease, specifically aortic stenosis, is also a well-known complication of renal dysfunction [38,39]. Gibson et al. [40] utilized eGFR in the analysis of outcomes after valve replacement. They showed that a decrease in eGFR of 10 mL/min/1.73 m^2^ corresponded to a 31% increase in the risk of postoperative death, and eGFR less than 60 mL/min/1.73 m^2^ was the most useful variable in predicting 30 day and midterm mortality. The postoperative GDF-15 value has also been reported to reflect post-operative acute kidney injury (AKI) in CABG patients [27]. In our study, using multivariate analysis, eGFR was an independent determinant of the preoperative GDF-15 level, even after adjusting for age, sex, and BMI. In addition, the GDF-15 level, was an independent determinant of eGFR. In ROC curve analysis, different cut-off levels of GDF-15 were used to predict eGFR lower than 60 mL/min/1.73 m^2^, and the AUCs was 92.1% with an optimal cut-off value of 1154 pg/mL. Given that the normal range for human serum GDF-15 level has been reported to be 150-1150 pg/mL [41], and 733-999 pg/mL [42], the cut-off value of 1154 pg/mL shown in our study appears to be a reasonable value for detecting CKD in patients hospitalized for cardiovascular surgery. Thus, GDF-15 appears to be a novel biomarker for identifying surgical risk in patients undergoing cardiovascular surgery. However, further studies using postoperative follow-up are needed to clarify whether the preoperative GDF-15 level can be used as a novel biomarker to identify surgical risk in patients undergoing cardiovascular surgery. 

### 4.2. Association of Serum GDF-15 Levels with Muscle Loss 

In cardiac surgery such as TAVI, frailty is identified as a major predictor of adverse outcomes in older surgical patients [1,3,4,5]. Sarcopenia, the skeletal muscle loss associated with aging, also includes low physical function (grip strength, walking speed) as a component of frailty [6,7]. Sarcopenia is frequently associated with chronic diseases including heart failure, COPD, CKD, cancer, wasting, and cachexia [7]. Thus, frailty and sarcopenia increase the risk of complications and mortality when cardiovascular surgery is performed. We have previously shown that the prevalence of sarcopenia including sarcopenic obesity in CVD patients was 47.5% in males and 60.2% in females [34]. In the present study using preoperative cardiovascular surgery patients, sarcopenia was identified in 36%. In these cardiovascular surgery patients, we found that the circulating GDF-15 level had a negative correlation with SMI, hand-grip strength, and anterior TMth. In multivariate analysis, the GDF-15 level was an independent determinant of anterior TMth, even after correction for age, gender, and BMI. In addition, preoperative GDF-15 levels were significantly higher in those with than without sarcopenia. The results were consistent with previous reports showing that the circulating GDF-15 concentration was negatively correlated with the cross-sectional area of rectus femoris in COPD [30], ICU [32], and cancer patients [33]. Bloch et al. [31] showed that patients who show wasting of the rectus femoris following cardiac surgery are exposed to a more sustained elevation of GDF-15 than those withour muscle wasting. Moreover, several recent papers reported the relationships between GDF-15 levels and physical function such as gait speed, and hand-grip strength in community-dwelling older adults [43,44]. In the present study, a significant relationship between hand-grip strength/knee extension and the GDF-15 level was observed in preoperative cardiovascular surgery patients. From these results, it is very likely that circulating preoperative GDF-15 level may be a biomarker for identifying muscle wasting and sarcopenia in cardiovascular surgery patients. 

Several mechanisms underlying GDF-15-induced muscle wasting have been proposed. First, an increase of GDF-15 may cause appetite loss, anorexia, and cachexia as shown in patients with cancer [41,45]. Animal studies also showed that GDF15 causes anorexia/cachexia, and then weight loss through a direct effect on the hypothalamus [46,47]. On the other hand, animal studies showed that local over-expression of GDF-15 in mice causes wasting of the tibialis anterior muscle directly [30], and the in vitro studies using C2C12 myotubes showed that GDF-15 treatment elevates expression of muscle atrophy-related genes, and causes muscle wasting, possibly by a direct effect of GDF-15 on skeletal muscle [32]. Further studies are required to clarify the mechanisms of action of GDF-15 on muscle wasting in patients with CVD including cardiovascular surgery.

### 4.3. Relationships between GDF-15 and TNFα, IGF-1 Concentration

It has been reported that several pro-inflammatory cytokines including TNFα are associated with cachexia and anorexia in both humans and rodents [48,49]. Higher concentrations of TNFα are also associated with a decline in muscle mass and grip strength in older persons [50]. In addition, systemic inflammation and increased circulating TNFα levels have been implicated in various conditions accompanied by muscle atrophy [51]. GDF-15 has been reported to be induced by inflammatory cytokines such as TNFα [10]. In our study, the serum GDF-15 level was positively correlated with the TNFα level, and a correlation was observed between TNFα and muscle mass and grip strength in men. However, multivariate regression analysis showed that that GDF-15, but not TNFα, was the independent determinant of anterior TMth (supine), even adjusting for age, sex, and BMI. These results suggest that GDF-15 is an independent marker of muscle mass, irrespective of TNF-α. Moreover, IGF-1 levels decrease with age and are regarded as a potential mediator of sarcopenia or frailty [52]. Wang et al. [53] showed that GDF-15 inhibits the release of IGF-1 from the liver in children with concomitant disease and failure. In our study, a negative correlation between GDF-15 and IGF-1 concentrations was observed only in females, but not in males. The reason of this sex discrepancy remains unknown. However, whereas the present study showed that IGF-1 was correlated negatively with age and positively with muscle mass, and anterior TMth, there were no significant differences in IGF-1 levels between patients with and without sarcopenia. Furthermore, we found no relationships between IGF-1 levels and hand-grip/extension strength. Thus, the contribution of serum IGF-1 levels to sarcopenia and muscle wasting remains unclear in our cardiovascular surgery patients. 

### 4.4. Limitations

This study has several limitations. First, the results do not imply causality, because it was a cross-sectional study. Second, the study had a small number of cardiovascular surgery patients, especially females, and the patients underwent different types of cardiovascular surgery. Therefore, our findings are not necessarily applicable to the general population of cardiovascular surgery patients. Furthermore, most of the subjects had medical treatment. The use of drugs such as β-blockers, ACE-I, and ARB might have affected serum cytokine level. Therefore, the further studies using a large number of patients are required to clarify the pathophysiological roles of GDF-15 in preoperative cardiovascular surgery patients. 

## 5. Conclusions

An elevated GDF-15 level reflects muscle wasting as well as renal dysfunction in preoperative cardiovascular surgery patients. Thus, serum GDF-15 concentration may be a novel biomarker for identifying high-risk patients with muscle wasting and renal dysfunction in cardiovascular surgery patients.

## Figures and Tables

**Figure 1 jcm-08-01576-f001:**
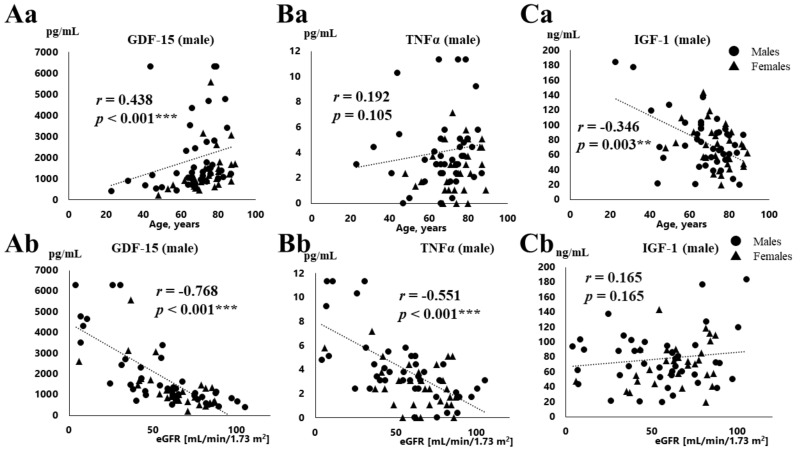
Correlations between clinical data (age, eGFR) and serum concentrations of GDF-15, TNF-α, and IGF-1. Relationships between laboratory data (age (**a**), eGFR (**b**)) and serum concentrations of GDF-15 (**Aa**,**Ab**), TNF-α (**Ba**,**Bb**) and IGF-1 (**Ca**,**Cb**) in males and females. ** *p* < 0.01, *** *p* < 0.001.

**Figure 2 jcm-08-01576-f002:**
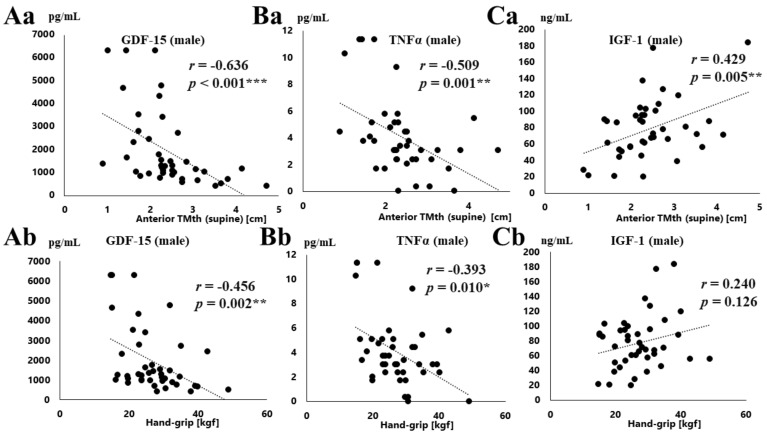
Correlations between the physical data (anterior thigh muscle thickness, grip strength) and serum concentrations of GDF-15, TNF-α, and IGF-1. Relationships between the laboratory data (anterior thigh muscle thickness (TMth, supine) (**a**), grip strength (**b**) and serum concentrations of GDF-15 (**Aa**,**Ab**), TNF-α (**Ba**,**Bb**) and IGF-1 (**Ca**,**Cb**) in males * *p* < 0.05, ** *p* < 0.01, *** *p* < 0.001.

**Figure 3 jcm-08-01576-f003:**
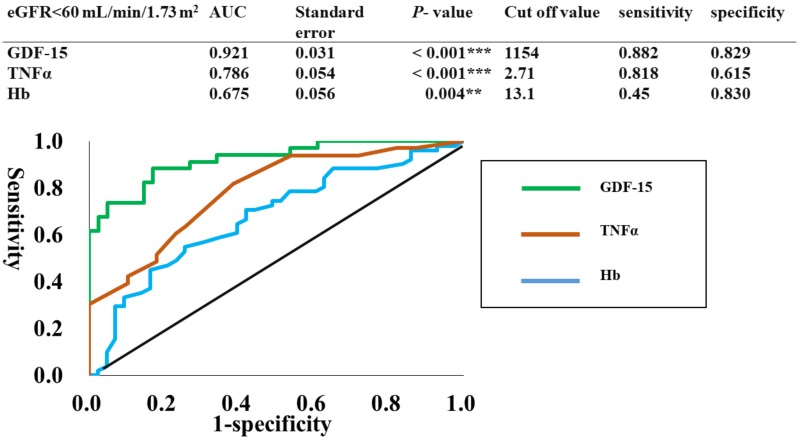
ROC curves to identify the optimal cut-off level of GDF-15, TNFα, and Hb for detecting eGFR < 60. In the ROC curves shown, different cut-off values of GDF-15, and TNFα, and Hb were used to predict eGFR < 60, with true positives plotted on the vertical axis (sensitivity) and Figure 1. plotted on the horizontal axis.

**Table 1 jcm-08-01576-t001:** Patient Characteristics.

Number	72
Male, Female	42, 30
Age, y	69.9 ± 13.1
BMI, kg/m^2^	24.3 ± 3.9
Risk factors (percentage)	
Hypertension	75
Diabetes	35
Dyslipidemia	46
Smoking	11
Hemodialysis	8
NYHA classification	2.2 ± 1.0
Coronary artery disease (percentage)	
0-vessel disease	53
1-vessel disease	11
2-vessel disease	6
3-vessel disease	30
Cardiovascular surgery (percentage)	
CABG	26
AVR	21
Other valve replacement/repair (MVR, MVP, TAP, TAR, LAAAC)	17
CABG combined with valve replacement/repair (AVR, MVP, TAP)	8
AVR combined with other valve (MVP, TAP, LAAC) or aortic diseases (TAR)	11
Aortic disease (TAR, TEVAR, et al.)	7
Others	10
Drugs (percentage)	
β-blockers	49
Ca-blockers	38
ACE-I/ARB	58
Diuretics	49
Statins	53
Oral antidiabetic drugs	31
Insulin	8

The mean ± SD values are shown. BMI, body mass index; NYHA, New York Heart Association; CABG, coronary artery bypass grafting; AVR, aortic valve replacement; MVR, mitral valve replacement; MVP, mitral valve plasty; TAP, tricuspid annuloplasty; LAAC, left atrial appendage closure; TAR, total arch replacement; TEVAR, thoracic endovascular aortic repair; ACE-I, angiotensin converting enzyme inhibitors; ARB, angiotensin II receptor blockers; Oral anti-diabetic drugs included α-glucosidase inhibitors, sulfonylurea, biguanide, dipeptidyl peptidase-4 inhibitors, and sodium glucose cotransporter 2 inhibitors.

**Table 2 jcm-08-01576-t002:** Sex differences in various parameters.

	Total (*n* = 72)	Male (*n* = 42)	Female (*n* = 30)
Age, years	69.9 (13.1)	66.9 (14.4)	73.7 (10.0) *
BMI, kg/m^2^	24.3 (3.9)	24.9 (4.5)	24.7 (5.5)
NYHA classification	2.2 (1.0)	2.3 (1.1)	2.1 (0.9)
Gait speed, m/s	0.93 (0.32)	0.99 (0.34)	0.86 (0.28)
Hand-grip strength, kgf	22.8 (8.5)	27.1 (7.9)	16.5 (4.6) ***
Knee extension strength, kgf	20.8 (9.5)	24.4 (9.3)	15.6 (7.1) ***
Body fat percentage, %	32.3 (9.3)	28.4 (7.8)	37.6 (8.7) ***
Skeletal muscle mass index (SMI), kg/m^2^	6.5 (1.4)	7.2 (1.3)	5.4 (0.9) ***
Anterior thigh muscle thickness (TMth) (supine), cm	2.28 (0.75)	2.41 (0.80)	2.1 (0.6)
Anterior thigh muscle thickness (TMth) (standing), cm	3.47 (0.95)	3.68 (0.97)	3.2 (0.9) *
HbA1c, %	6.2 (0.9)	6.3 (1.0)	6.0 (0.8)
BNP, pg/mL	355 (570)	399 (673)	268 (345)
eGFR, ml/min/1.73 m^2^	58.2 (24.0)	55.7 (26.6)	63.4 (19.3)
Hb, g/dL	12.2 (1.8)	12.3 (1.9)	11.9 (1.7)
HOMA-IR	2.75 (4.20)	3.46 (5.29)	1.64 (1.29)
hsCRP, mg/L	5.9 (12)	7.4 (13.8)	3.3 (7.9)
GDF-15, pg/mL	1676 (1465)	1928 (1655)	1325 (1078)
TNFα, pg/mL	3.5 (2.8)	4.1 (2.9)	2.6 (1.9) *
IGF-1, ng/mL	74.4 (33.4)	77.3 (36.5)	70.4 (28.5)

* *p* < 0.05. *** *p* < 0.001. Males vs. Females hsCRP, high sensitivity C-reactive protein; BNP, brain natriuretic peptide; eGFR, estimate glomerular filtration rate; HOMA-IR, Homeostasis model assessment of insulin resistance; GDF-15, growth differentiation factor-15; TNFα, tissue necrosis factor α; IGF-1, insulin growth factor-1.

**Table 3 jcm-08-01576-t003:** Correlation matrix between various parameters and serum GDF-15, and TNFα, IGF-1 concentration.

	GDF-15 Males/Females	TNFα Males/Females	IGF-1 Males/Females
Age	0.436 (0.004) **/0.637 (<0.001) ***	0.244 (0.120)/0.261 (0.164)	−0.319 (0.036) */−0.339 (0.066)
BMI	−0.143 (0.367)/−0.062 (0.745)	−0.263 (0.092)/−0.055 (0.772)	0.047 (0.769)/0.194 (0.303)
HbA1C	−0.155 (0.340)/−0.229 (0.223)	−0.154 (0.342)/−0.127 (0.503)	0.261 (0.104)/0.344 (0.063)
BNP	0.427 (0.005) **/0.480 (0.007) **	0.465 (0.002) **/0.214 (0.257)	−0.059 (0.710)/−0.353 (0.055)
eGFR	−0.792 (<0.001) ***/−0.726 (<0.001) ***	−0.642 (<0.001) ***/−0.394 (0.031) *	0.144 (0.363)/0.301 (0.106)
Hb	−0.560 (<0.001) ***/−0.370 (0.044) *	−0.566 (<0.001) ***/−0.065 (0.732)	0.079 (0.617)/−0.006 (0.977)
Body fat percentage	−0.140 (0.403)/0.300 (0.128)	−0.146 (0.382)/0.135 (0.502)	−0.119 (0.479)/0.197 (0.326)
SMI	−0.392 (0.014) */−0.529 (0.005) **	−0.368 (0.021) */−0.189 (0.346)	0.313 (0.053)/0.153 (0.446)
Hand-grip	−0.456 (0.002) **/−0.656 (<0.001) ***	−0.393 (0.010) */−0.298 (0.117)	0.240 (0.126)/0.244 (0.202)
Knee extension	−0.222 (0.169)/−0.541 (0.003) **	−0.431 (0.005) **/−0.192 (0.329)	0.140 (0.390)/0.061 (0.758)
Gait speed	−0.218 (0.165)/−0.558 (0.002) **	−0.190 (0.229)/−0.336 (0.074)	0.256 (0.102)/0.253 (0.186)
Anterior TMth (supine)	−0.636 (<0.001) ***/−0.391 (0.044) *	−0.509 (0.001) **/−0.200 (0.316)	0.429 (0.005) **/0.057 (0.779)
Anterior TMth (standing)	−0.600 (<0.001) ***/−0.557 (0.003) **	−0.434 (0.005) **/−0.267 (0.178)	0.366 (0.020) */0.301 (0.127)
GDF-15	-/-	0.657 (<0.001) ***/0.434 (0.017) *	−0.203 (0.198)/−0.553 (0.002) **
TNFα	0.657 (<0.001) ***/0.434 (0.017) *	-/-	−0.233 (0.137)/−0.479 (0.007) **

* *p* < 0.05 ** *p* < 0.01 *** *p* < 0.001. SMI, skeletal muscle mass index; TMth, thigh muscle thickness.

**Table 4 jcm-08-01576-t004:** Multiple linear regression analysis between serum GDF-15 levels and the clinical parameters.

**A: Multiple linear regression analysis of GDF15 and the clinical data**				
	Dependent variable: log (GDF−15)			
	Model 1	Model 2	Model 3	Model 4
Independent variable.	β-value (*p*)	β-value (*p*)	β-value (*p*)	β-value (*p*)
eGFR	−0.650 (<0.001) ***	−0.655 (<0.001) ***	−0.613 (<0.001) ***	−0.597 (<0.001) ***
BNP (log)	−0.009 (0.929)	−0.011 (0.912)	−0.042 (0.649)	−0.040 (0.663)
Hb	−0.106 (0.257)	−0.110 (0.253)	−0.079 (0.372)	−0.084 (0.343)
SMI	0.036 (0.722)	0.031 (0.771)	−0.171 (0.139)	−0.196 (0.098)
Hand−grip strength	0.106 (0.312)	0.323 (0.104)	−0.105 (0.367)	−0.059 (0.632)
Anterior TMth (supine)	−0.358 (0.001) **	−0.362 (0.001) **	−0.233 (0.033) *	−0.272 (0.019) *
**B: Multiple linear regression analysis of anterior thigh muscle thickness (supine) and serum markers**				
	Dependent variable: anterior thigh muscle thickness (supine)			
	Model 1	Model 2	Model 3	Model 4
Independent variable	β-value (*p*)	β-value (*p*)	β-value (*p*)	β-value (*p*)
GDF−15 (log)	−0.401 (0.005) **	−0.311 (0.024) *	−0.384 (0.007) **	−0.390 (0.004) **
TNFα (log)	−0.007 (0.955)	−0.054 (0.671)	−0.068 (0.584)	−0.054 (0.644)
IGF−1	0.256 (0.031) *	0.094 (0.456)	0.071 (0.656)	0.078 (0.506)
**C: Multiple linear regression analysis of eGFR and the clinical data**				
	Dependent variable: eGFR			
	Model 1	Model 2	Model 3	Model 4
Independent variable	β-value (*p*)	β-value (*p*)	β-value (*p*)	β-value (*p*)
BNP (log)	−0.164 (0.070)	−0.149 (0.106)	−0.149 (0.113)	−0.170 (0.070)
hsCRP (log)	−0.070 (0.377)	−0.094 (0.262)	−0.097 (0.257)	−0.085 (0.999)
Hb	0.035 (0.685)	0.005 (0.958)	0.002 (0.984)	0.011 (0.657)
GDF−15 (log)	−0.583 (<0.001) ***	−0.571 (<0.001) ***	−0.577 (<0.001) ***	−0.565 (<0.001) ***
TNFα (log)	−0.177 (0.069)	−0.196 (0.050)	−0.198 (0.050)	−0.197 (0.050)

Model 1, unadjusted; Model 2, adjusted by age; Model 3, adjusted by age and sex; Model 4, adjusted by age, sex, and BMI.

**Table 5 jcm-08-01576-t005:** Differences in clinical data between the patients with and without sarcopenia.

	Male		Female	
	Sarcopenia (−)	Sarcopenia (+)	Sarcopenia (−)	Sarcopenia (+)
Number	28	11	14	13
Age (years)	63.4 (14.2)	75.6 (10.8) ***	68.9 (10.5)	76.2 (7.3) *
BMI (kg/m^2^)	26.0 (4.8)	22.6 (3.0) *	24.9 (3.1)	25.9 (7.3)
Physical capacity				
Gait speed (m/s)	1.08 (0.34)	0.77 (0.28) *	1.01 (0.17)	0.73 (0.32) *
Grip strength (kgf)	30.0 (7.4)	20.2 (4.4) ***	20.1 (3.1)	13.9 (3.5) ***
Knee extension (kgf)	26.9 (8.4)	17.5 (7.3) **	19.2 (7.5)	13.6 (4.8)*
BIA findings				
Body fat percentage (%)	28.8 (7.4)	27.9 (8.9)	35.7 (6.6)	39.8 (10.3)
Skeletal muscle mass index (SMI) (kg/m^2^)	7.68 (1.15)	5.90 (0.53) ***	6.05 (0.53)	4.74 (0.63) ***
Muscle thickness				
Anterior TMth (supine) (cm)	2.64 (0.68)	1.65 (0.53) ***	2.45 (0.68)	1.89 (0.48) *
Anterior TMth (standing) (cm)	3.98 (0.88)	2.78 (0.46) ***	3.70 (0.74)	2.77 (0.79) **
HbA1c, %	6.3 (0.9)	6.4 (1.3)	6.12 (0.98)	5.97 (0.60)
BNP, pg/mL	209 (285)	646 (722) **	162 (250)	366 (427) *
eGFR, ml/min/1.73 m^2^	59.7 (26.4)	47.9 (26.1)	70.6 (14.4)	56.1 (22.1)
Hb, g/dL	13.0 (1.6)	11.0 (1.7)	12.5 (1.7)	11.5 (1.6)
hsCRP, mg/L	7.1 (12.7)	9.9 (18.5)	4.5 (11.3)	2.0 (2.3)
GDF-15, pg/mL	1483 (1125)	3053 (2346) *	891 (700)	1625 (1302) *
TNFα, pg/mL	3.06 (1.95)	6.07 (3.27) **	2.23 (2.10)	2.92 (1.66)
IGF-1, ng/mL	84.1 (40.0)	62.7 (25.6)	74.7 (23.6)	70.6 (34.6)

* *p* < 0.05. ** *p* < 0.01***. *p* < 0.001. Males vs. Females.

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
