# Peer review of "Growth Differentiation Factor-15 (GDF-15) Is a Biomarker of Muscle Wasting and Renal Dysfunction in Preoperative Cardiovascular Surgery Patients"

_jcm, 2019, doi:10.3390/jcm8101576_

Round 1
Reviewer 1 Report
Nakajima et al. evaluated the potential of GDF-15 as biomarker of muscle wasting and renal dysfunction in preoperative cardiovascular surgery patients. The topic is of interest. Nevertheless, there are a number of conceptual and methodological limitations that preclude the publication of the manuscript.
The authors did not investigated the pathophysiological role of GDF-15. The experimental design does not allow to explore the causal role of GDF-15. The manuscript is based on patients, the mechanism has not been explored
The first part of the introduction is mainly based on frailty. However, patients are not classified in based on frailty characteristics. Sarcopenia is one of the factors that participate in frailty. Authors could use one of the proposed classification for frailty.
Other biomarkers should be included, at least NT-proBNP, and troponins I and T.
Blood collection protocol is unclear. The inclusion of TNF-alpha and IGF-1 in the study is not justified.
There are several paragraphs in methods section that should be included in results section (line 123 and line 128).
Statistical analysis section is poor. A biomarker study required a deep characterization. Mann-Whitney test does not compare means. The use of Kolmogorov-Smirnov test is unacceptable. Clinical, laboratory and physical data should also be included in the analyses.
In results, the comparison of males and females is unclear. Why the authors did performed this analysis?
There are results in the text that is already included in the tables.
Information contained in Table 1 and Table 2 is not the same.
Line 200: TNF-alpha?
There is a problem the Figures. Please, check the numbers.
Use P <0.001 instead P = 0.000
Line 264: Linear regression is not the same than correlation
The discussion is too large and hypothetical.
Reviewer 2 Report
This is a well designed study and I have only minor comments:
table 1: could authors provide percentages and not only number hs-CRP should be in mg/L and not in mg/dl (please correct units) authors should discuss their results and refer their GDF-15 concentrations to the recent study: Clin Chem Lab Med. 2019 Jun 26;57(7):1035-1043. doi: 10.1515/cclm-2018-0908.
Reviewer 3 Report
The authors studied the interest of GDF-15 on pre operative assessment of patients before cardiovascular surgery. The results are interesting and the clinical background is well described
I would be interested to have the incidence and predictors of acute kidney injury after the surgery, and to confirm in this cohort that pre operative GDF-15 levels are predictive of AKI
I feel that the discussion would benefit to be organized in sub chapters to improve the readability
Overall, i congratulate the authors for their thorough analysis of this population and to add another piece in the understanding of GDF-15
Round 2
Reviewer 1 Report
The manuscript has been improved